# Pre-analytical error for three point of care venous blood testing platforms in acute ambulatory settings: A mixed methods service evaluation

**Thomas R. Fanshawe** [1]*, **Margaret Glogowska** [1], **George Edwards** [1], **Philip J. Turner** [1], **Ian Smith** [2], **Rosie Steele** [2], **Caroline Croxson** [3], **Jordan S. T. Bowen** [2], **Gail N. Hayward** [1]

**1** Nuffield Department of Primary Care Health Sciences, University of Oxford, Oxford, United Kingdom, **2** Oxford University Hospitals NHS Foundation Trust, Oxford, United Kingdom, **3** University of Oxford, Oxford, United Kingdom

* thomas.fanshawe@phc.ox.ac.uk

**Data Availability Statement:** All relevant data are within the paper and its Supporting Information files.

## Abstract

### Introduction

Point of care blood testing to aid diagnosis is becoming increasingly common in acute ambulatory settings and enables timely investigation of a range of diagnostic markers. However, this testing allows scope for errors in the pre-analytical phase, which depends on the operator handling and transferring specimens correctly. The extent and nature of these pre-analytical errors in clinical settings has not been widely reported.

### Methods

We carried out a convergent parallel mixed-methods service evaluation to investigate pre-analytical errors leading to a machine error reports in a large acute hospital trust in the UK. The quantitative component comprised a retrospective analysis of all recorded error codes from Abbott Point of Care i-STAT 1, i-STAT Alinity and Abbott Rapid Diagnostics Afinion devices to summarise the error frequencies and reasons for error, focusing on those attributable to the operator. The qualitative component included a prospective ethnographic study and a secondary analysis of an existing ethnographic dataset, based in hospital-based ambulatory care and community ambulatory care respectively.

### Results

The i-STAT had the highest usage (113,266 tests, January 2016-December 2018). As a percentage of all tests attempted, its device-recorded overall error rate was 6.8% (95% confidence interval 6.6% to 6.9%), and in the period when reliable data could be obtained, the operator-attributable error rate was 2.3% (2.2% to 2.4%). Staff identified that the most difficult step was the filling of cartridges, but that this could be improved through practice, with a perception that cartridge wastage through errors was rare.

**Funding:** This research was funded by the National Institute for Health Research (NIHR) Community Healthcare MedTech and In Vitro Diagnostics Co-operative at Oxford Health NHS Foundation Trust. The views expressed are those of the authors and not necessarily those of the NHS, the NIHR or the Department of Health and Social Care. The project was also supported by a grant from Becton, Dickinson and Company. The funders had no role in study design, data collection and analysis, decision to publish, or preparation of the manuscript.

**Competing interests:** The project was supported by a grant from a commercial source (Becton, Dickinson and Company). The funders had no role in study design, data collection and analysis, decision to publish, or preparation of the manuscript. This does not alter our adherence to PLOS One policies on sharing data and materials.

## Conclusions

In the observed settings, the rate of errors attributable to operators of the primary point of care device was less than 1 in 40. In some cases, errors may lead to a small increase in resource use or time required so adequate staff training is necessary to prevent adverse impact on patient care.

## Introduction

Point of care testing (POCT) has gradually grown in popularity, both in primary care, where it is argued to be a key element of antimicrobial stewardship [1], and in secondary and tertiary care as new technologies emerge [2–4]. Although analytical performance of POCT platforms is available in manufacturers' technical documentation, the frequency and nature of errors that may arise when using these platforms in the clinical environment is less commonly reported, and definitions of pre-analytical error are wide-ranging and inconsistently applied [5].

In practice, POCT may be performed by multiple operators with varying levels of experience [6]. This makes errors that may be attributable to the user in the pre-analytical phase of testing, such as incorrectly filling or mishandling a cartridge, especially relevant. There is a concern that the risk of error using POCT in clinical settings may be higher than in the laboratory [7], although even in the laboratory environment, rates of pre-analytical error may be non-negligible [8–10].

A study in secondary care in the UK estimated an overall error rate from a variety of POCT devices as less than 1% of total tests performed, with 32% of all errors occurring in the pre-analytical phase and the majority of these having only minor impact on patient outcomes, such as delayed diagnosis [11]. However, these rates were based on errors logged individually by the operators rather than automatically recorded.

There is therefore limited published evidence for the frequency of, and reasons for, pre-analytical errors when using POCT devices in ambulatory settings. The aims of this mixed-methods study, set in a large acute hospital trust in the UK, are to quantify error rates by error category, to identify patterns of errors by hospital department and over time, and to explore the processes involved in POCT and possible reasons underlying pre-analytical error using qualitative, ethnographic research.

## Materials and methods

The mixed methods design used was convergent parallel [12]. The quantitative and qualitative components were conducted and analysed separately, and the findings from both components were then integrated, to provide a more comprehensive picture of pre-analytical error in this setting.

### Quantitative

**Setting.**    The quantitative component uses routinely collected error code information reported by POCT blood testing platforms in use at Oxford University Hospitals NHS Foundation Trust (OUHFT) between January 2016 and December 2018, namely the Abbott i-STAT [12, 13], Abbott i-STAT Alinity [14], and Afinion AS100 [15] and Afinion 2 [16] analyzers. More than 40 settings were classified by location and, where applicable, department, ward or care unit. These reflect the wide scope in which POCT is deployed, including in particular

**Table 1. Tests available on POCT platforms in use during study period.**

| Device | Cartridge | Qualitative (secondary analysis, 2014–15) | Qualitative (primary study, Jan-Feb 2019) | Quantitative |
|---|---|:---:|:---:|:---:|
| i-STAT | ACT | | | Jan 2016 –Dec 2018 |
| | BNP | | | |
| | CG4+ | ✓ | ✓ | |
| | CHEM8+ | ✓ | ✓ | |
| | Creatinine (Crea) | | | |
| | E3+ | | | |
| | Glucose (G) | | | |
| | β-hCG | | | |
| | PT/INR | ✓ | | |
| | Troponin (cTnI) | ✓ | | |
| i-STAT Alinity | CG4+ | | | Apr 2017 –Dec 2018 |
| | CHEM8+ | | | |
| | Creatinine (Crea) | | | |
| Afinion | CRP | ✓ | | Jul–Dec 2018 |
| | HbA1c | | | |
| | Lipids | | | |

emergency departments and acute ambulatory units, but also pre-operative assessment units, departments providing medical imaging services, and community hospitals. Medical specialties in which POCT is carried out are also wide-ranging and include radiology, cardiology, gynaecology, renal medicine and paediatrics. In most locations, multiple healthcare staff performed POCT, but reliable user-level frequency data were unavailable. Tests used and relevant timeframes are shown in Table 1.

A large majority of POCT in OUHFT uses venous blood samples and so results presented here are for venous samples only. Radiometer blood gas analyzers [17] are also in widespread use in OUHFT but are not considered further here as a comprehensive report of error codes could not be obtained retrospectively.

**Data management.** POCT platforms in OUHFT are connected to a POCcelerator Data Management system (Siemens Healthineers [18]), overseen by the Department of Clinical Biochemistry. Data were extracted using the Structured Query Language Report and Statistics Module. POCT information was obtained by cross-referencing a 'log extract' data file with a 'workload' data file. The log extract file contains every instance in which the device returned an error, with the machine and location ID, a time stamp recorded to the nearest minute, and the error code or code category. Information messages and messages relating to quality control were removed before analysis. Information about which analyte was being tested when errors occurred is not available. The workload data file lists every point of care (POC) test for which a valid result was returned: the machine and location ID, a time stamp, and the name of the analyte tested or cartridge used, but not the test result itself. Comparison of these files allows error rates to be estimated.

Individual POCT machines frequently moved between clinical locations and often had intermittent periods out of operation. Data from each machine were therefore restricted to periods for which both reliable workload and error data were available. For locations using more than one machine, data from each machine were aggregated so that results are presented at a per-location level, consistent with our focus being the assessment of operator errors rather than errors attributable to the particular analyzer. Duplications with identical time stamp, location and error code were removed.

**Error code classification.** We follow the taxonomy suggested by Meier & Jones [19], following Kost [20]. Our focus is on the 'specimen collection' and 'specimen evaluation' stages of the pre-analytical phase, while recognising that in some circumstances errors reported as occurring in the analytic phase may also have been caused by incorrect specimen collection. Error codes are subsetted to attribute failures likely to be due to user error, such as overfilling or underfilling the cartridge, and those that are outside the influence of the user, such as environmental factors and analyzer malfunction.

For the i-STAT between February 2017 and December 2018, error codes taken from the i-STAT technical manual [21] that occurred at least once in the study dataset were grouped into categories of comparable codes (S1 Appendix), and further grouped into those deemed likely to correspond to operator errors resulting in cartridge wastage. This categorisation is not definitive as some codes could plausibly correspond to either an operator error or a non-operator error. Before February 2017, the log extract record did not contain specific i-STAT error codes but instead grouped codes into eight broader categories (S1 Appendix). Descriptive labels applied to these categories were inconsistent, with a preponderance of codes being assigned to a catch-all 'Other' category, so error rate information for the i-STAT is reported separately for January 2016-February 2017 and February 2017-December 2018.

Numerical i-STAT Alinity error codes were categorised based on the technical manual [22] in terms of their cause and suggested resolution, and a similar procedure was followed for the Afinion [23] (S1 Appendix).

**Statistical analysis.** Monthly error rates were estimated as the number of recorded errors carrying the appropriate error code(s) divided by the total workload. The latter was estimated as the sum of the number of recorded errors and the number of recorded uses from which a valid test result was obtained.

For example, an attempt that produced an error code corresponding to an underfilled cartridge, followed by a second correctly-performed attempt on the same patient, would contribute one recorded error and two test attempts (workload). An attempted test that led to a device failure and no subsequent test being successfully performed would contribute one recorded error and one test attempt. It was not possible to determine with certainty whether any individual recorded error resulted in wastage of a cartridge as opposed to reinsertion of the same cartridge.

Monthly error frequencies and rates, with 95% confidence intervals (CIs), are expressed in tabular form and graphically to show trends over time. Results are also presented by error code category.

For the i-STAT, information is also presented to illustrate the between-location (e.g. hospital department or ward) variation in error rates, using numerical and graphical summaries such as stacked bar charts and forest plots. Pearson's correlation was used to quantify the association between location-specific error rate and total workload. Details of specific locations are not disclosed. Statistical analysis used R version 3.5.2 [24].

**Ethics and Information Governance.** As this retrospective evaluation of routinely-collected data does not use personally-identifiable patient information, ethics committee approval was not mandated. The project was prospectively approved (November 2018) as an OUHFT service evaluation (Datix reference number CSS-BIO-4 5315).

## Qualitative

**Setting.** There were two strands to the qualitative component of this study. The first involved a secondary analysis of a primary qualitative ethnographic dataset. It was conducted in a community-based geriatrician-led ambulatory care setting aimed at frail and older

patients in Oxford Health NHS Foundation Trust (OHFT) between 2014 and 2015, to explore how POCT had become embedded in working practice [25].

The second strand comprised a prospective ethnographic study which followed on from the secondary analysis and took place in consultant-delivered acute ambulatory care at OUHFT. The POCT platforms used in these settings are shown in Table 1.

**Data collection.** The dataset used for the secondary analysis comprised ethnographic field notes of observations (14 sets) and transcripts of semi-structured interviews with clinical staff using POCT (six interviews). The observations encompassed 14 occasions of venous blood sampling for POCT, resulting in the use of 27 cartridges– 15 CHEM8+, 10 CRP, one troponin and one INR. Blood sampling was observed on two occasions where blood could not be obtained.

The prospective ethnographic field work was conducted in January and February 2019. It comprised three visits (and corresponding sets of field notes), encompassing seven occasions of venous blood sampling, resulting in nine cartridges being used–three CHEM8+ and six CG4+.

**Data analysis.** The purpose of the secondary analysis was to investigate an existing ethnographic dataset, with particular focus on workflow issues and pre-analytical errors in carrying out the POC tests in an ambulatory care setting. The data in the primary study had been analysed thematically. A coding scheme had been developed which included both *a priori* items and emerging ideas. The data had been systematically coded using NVivo 10 and codes had been combined into broader categories and themes.

The analytic techniques used in the secondary analysis were similar to those applied in the primary analysis as recommended in the methodological literature [26]. This included reading and re-reading field notes and transcripts, open coding, with codes grouped into categories and themes. For the secondary analysis, a coding scheme was developed which included both the issues around workflow and pre-analytical errors and emerging ideas. The data were systematically coded using NVivo 11. The codes were combined into broader categories and themes.

The data collected in the prospective ethnographic strand of the qualitative component were also analysed thematically. A coding scheme was developed which included *a priori* items which had resulted from the secondary analysis of the original ethnographic dataset, areas of interest put forward by the research team for follow-up in the prospective ethnography, and emerging ideas. The data were systematically coded using NVivo 11 and codes were combined into broader categories and thematic areas.

**Ethics and Information Governance.** In the primary study, written informed consent was obtained from all interview participants in which they agreed that information about them could be stored in a way that meant they could not be identified so that the information could be used for research in the future. The study was approved by the University of Oxford Central University Research Ethics Committee (reference MSD-IDREC-C1-2014-113) and the relevant NHS Trust.

The prospective qualitative component was part of the service evaluation previously described. Permissions were sought for the qualitative researcher to be given access to the OUHFT settings where POCT was practised. Staff were made aware of the reason for the observation and all involved in POCT agreed to be observed. In all the instances which brought the observer into contact with patients, verbal consent from patients was sought first.

## Results

### Quantitative

**i-STAT.** Across the full study period (January 2016-December 2018) and all locations, 113266 i-STAT tests were performed. The most commonly used test cartridges were Troponin,

CHEM8+, Creatinine and CG4+, which together accounted for 88% of total workload (Table 2). Overall, there were 105578 returned results and 7688 errors, an error rate of 6.8% (95% CI 6.6% to 6.9%). This error rate was slightly higher in January 2016-February 2017 (2584/35113, 7.4% (7.1% to 7.6%)) than in February 2017-December 2018 (5104/78153, 6.5% (6.4% to 6.7%)).

Table 3 shows a breakdown of errors by error category (February 2017-December 2018). Of 5104 recorded errors, 1783 were assigned to user error categories (2.3% (95% CI 2.2% to 2.4%)). Overfilling occurred on 573 occasions (0.7%) and underfilling on 709 occasions (0.9%). In January 2016-February 2017, there were 2584 recorded errors, of which 694 (2.0% (95% CI 1.8% to 2.1%)) fell into the categories of underfilling, overfilling or insufficient sample.

Fig 1 shows the number of recorded results, user errors and non-user errors by location, for the 15 locations with the highest overall usage (more than 1000 tests in total). Although details of specific locations cannot be disclosed, emergency departments and acute ambulatory settings were among those with high frequencies of POCT. Error rates accounted for only a small proportion of total uses, with some variation between locations in the numbers falling into different user error categories such as underfill and overfill.

Fig 2 compares the estimated error rates and user error rates by location. Estimated error rates ranged between around 4% and 10% and user error rates were similar across locations, with most lying between 1% and 3%.

Over time, there was a steady increase in i-STAT use until September 2018 owing to the gradual introduction of machines into new locations, with a decline from September 2018, pending the introduction of a high sensitivity cardiac troponin assay. Error rates declined to around 7% during the second half of 2017, followed by a steeper decline to 4–5% towards the end of 2018 (S1 Graph).

The number of tests performed by location was not closely associated with the overall error rate (Pearson's $r = -0.11$, Fig 3), nor with the user-attributed error rate ($r = -0.17$).

## i-STAT Alinity

i-STAT Alinity workload and error data were available from six locations (April 2017-December 2018). There were 1829 recorded uses, comprising 1661 returned results and 168 errors (9.2%, 95% CI 7.9% to 10.6%), of which 95 (5.2%, 95% CI 4.2% to 6.3%) were classified as likely user errors. The breakdown of these codes by error category is shown in Table 4. The largest contributing component was Cause 3, encompassing a range of possible user errors primarily relating to cartridge handling.

**Table 2. Total test cartridge use, i-STAT.**

| Cartridge | Total workload (%) |
|---|---|
| ACT | 4325 (4.1%) |
| BNP | 1713 (1.6%) |
| CG4+ | 10494 (9.9%) |
| CHEM8+ | 22695 (21%) |
| Creatinine (Crea) | 14260 (14%) |
| E3+ | 136 (0.1%) |
| Glucose (G) | 935 (0.9%) |
| β-hCG | 2489 (2.4%) |
| PT/INR | 2902 (2.7%) |
| Troponin (cTnI) | 45269 (43%) |

**Table 3. Total frequencies and error rates of error code categories, i-STAT, February 2017–December 2018.**

| Error category | Frequency (%) | Likely user error | Total frequency (%) |
|---|---|---|---|
| Overfill | 573 (0.7%) | ✓ | 1783 (2.3%) |
| Underfill | 709 (0.9%) | ✓ | |
| Insufficient sample | 207 (0.3%) | ✓ | |
| Insertion error | 8 (0.01%) | ✓ | |
| Immunoassay–poor filling | 286 (0.4%) | ✓ | |
| Battery error | 465 (0.6%) | | 3321 (4.2%) |
| Barcode error | 41 (0.05%) | | |
| Coagulation error | 177 (0.2%) | | |
| Cartridge error | 1284 (1.6%) | | |
| Contact error | 394 (0.5%) | | |
| Analyzer/motor error | 522 (0.7%) | | |
| Immunoassay–analysis fluid | 130 (0.2%) | | |
| Immunoassay–error in QC run | 112 (0.1%) | | |
| Immunoassay–atypical data stream | 159 (0.2%) | | |
| Field length | 37 (0.05%) | | |
| TOTAL | 5104 (6.5%) | | 5104 (6.5%) |

Percentages are based on a total denominator of 78,153 tests. Refer to S1 Appendix for error category descriptors.

**Afinion.** Reliable data could be obtained from three locations in which the Afinion was in operation for CRP testing (July-December 2018). One of these was in ambulatory care and two in community hospitals. During this period there were 556 recorded uses, comprising 377 returned results and 179 errors (32% (95% CI 28% to 36%)), including 50 probable user errors (9.0% (95% CI 6.8% to 11.8%)).

Table 5 shows the breakdown of recorded errors by error code. The majority of probable user errors were attributed to insufficient sample volume (code 201), and more than half of these occurred at a single location.

## Qualitative

From the secondary analysis we built up a step-by-step picture of the whole process involved in venous blood sampling for POCT in a community-based ambulatory care setting. In the follow-up ethnography, we examined the process in a different setting, intentionally focusing on the factors which potentially contributed to pre-analytical errors. There were no discrepancies between the findings which derived from the secondary analysis and those from the follow-up ethnography. Our findings show that, across both settings, staff were aware of ways in which errors might occur in the blood sampling and POCT process. Their main concerns were difficulties with cartridge-based testing, difficulties with obtaining blood samples and accessing POCT devices. Staff were taking active steps to minimise the impact of these difficulties.

**Staff awareness of factors underlying pre-analytical error.** Staff perceived that the main difficulty they encountered, and an important contributor to pre-analytical error in cartridge-based POCT, was filling the i-STAT cartridges with the right amount of blood for testing:

"I mean the hardest bit is actually just getting the blood into the actual cartridge" [Interview 4, HCA, secondary analysis]

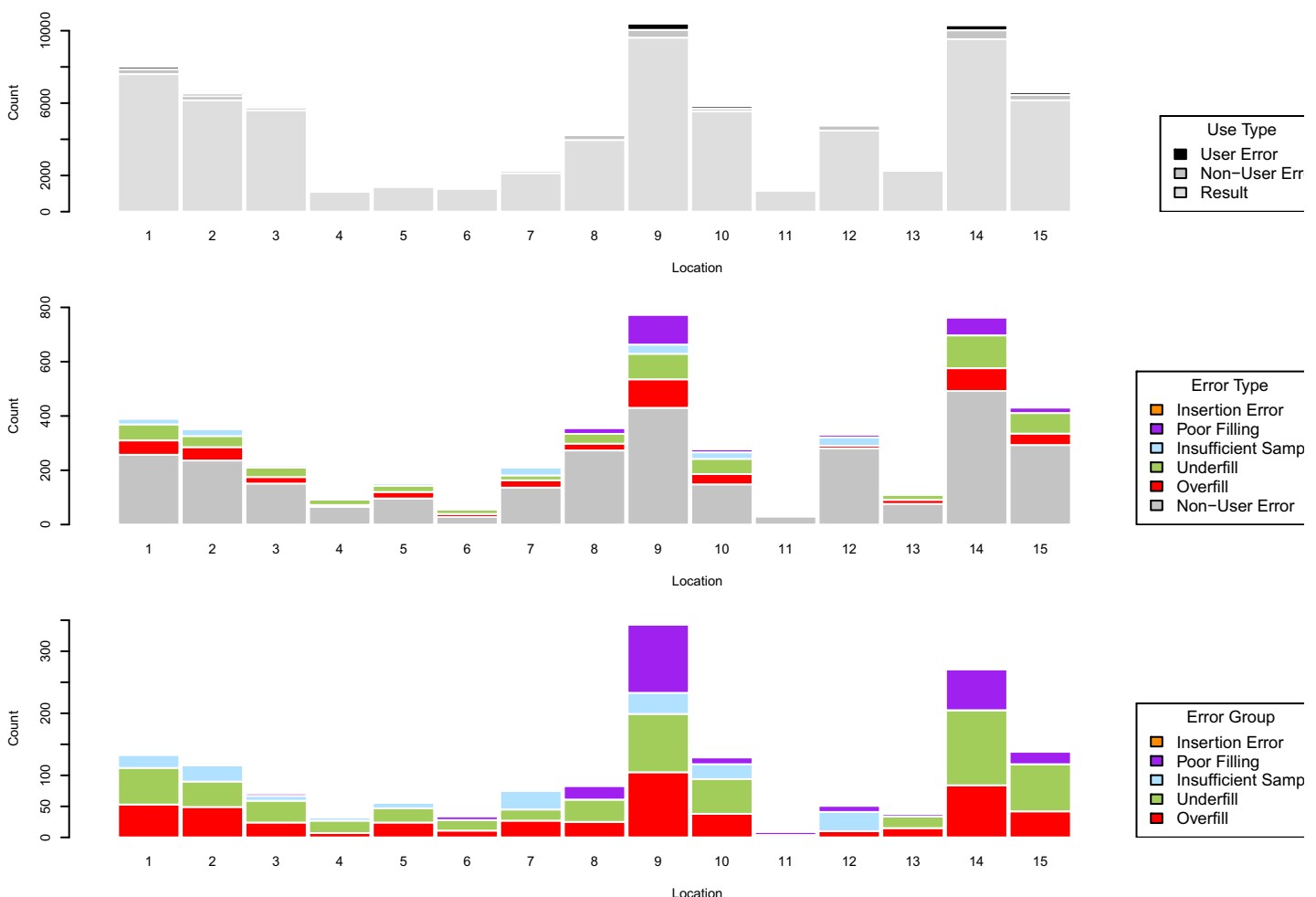

**Fig 1. i-STAT error rates in 15 locations with highest workload.** Total numbers of returned results, user errors and non-user errors (upper panel); breakdown of all errors by selected error category (middle panel); breakdown of user errors by selected error category (lower panel). In the upper panel, numbers of errors in some locations are too small to be easily visible.

One nurse described the process as 'fiddly' and sometimes 'messy', with the cartridges 'temperamental':

"...it is quite fiddly and sometimes the cartridges are quite hard to fill...they're, sometimes they're like a sort of little bit temperamental...you know, for example, that's the fiddly part is filling up the cartridge and that's what everybody gets a bit kerfuffled about and sometimes it goes everywhere and it's a bit messy." [Interview 3, Nurse, secondary analysis]

The 'fiddliness' of the i-STAT cartridges was contrasted with the CRP one:

"The CRP cartridge stops when it has the exact amount of blood it needs–this is better than the i-STAT cartridges, it would be good if all the cartridges were like that." [Interview 2, HCA, notes, secondary analysis]

"CRP is really easy to use because it's got like a little straw." [Interview 3, Nurse, secondary analysis]

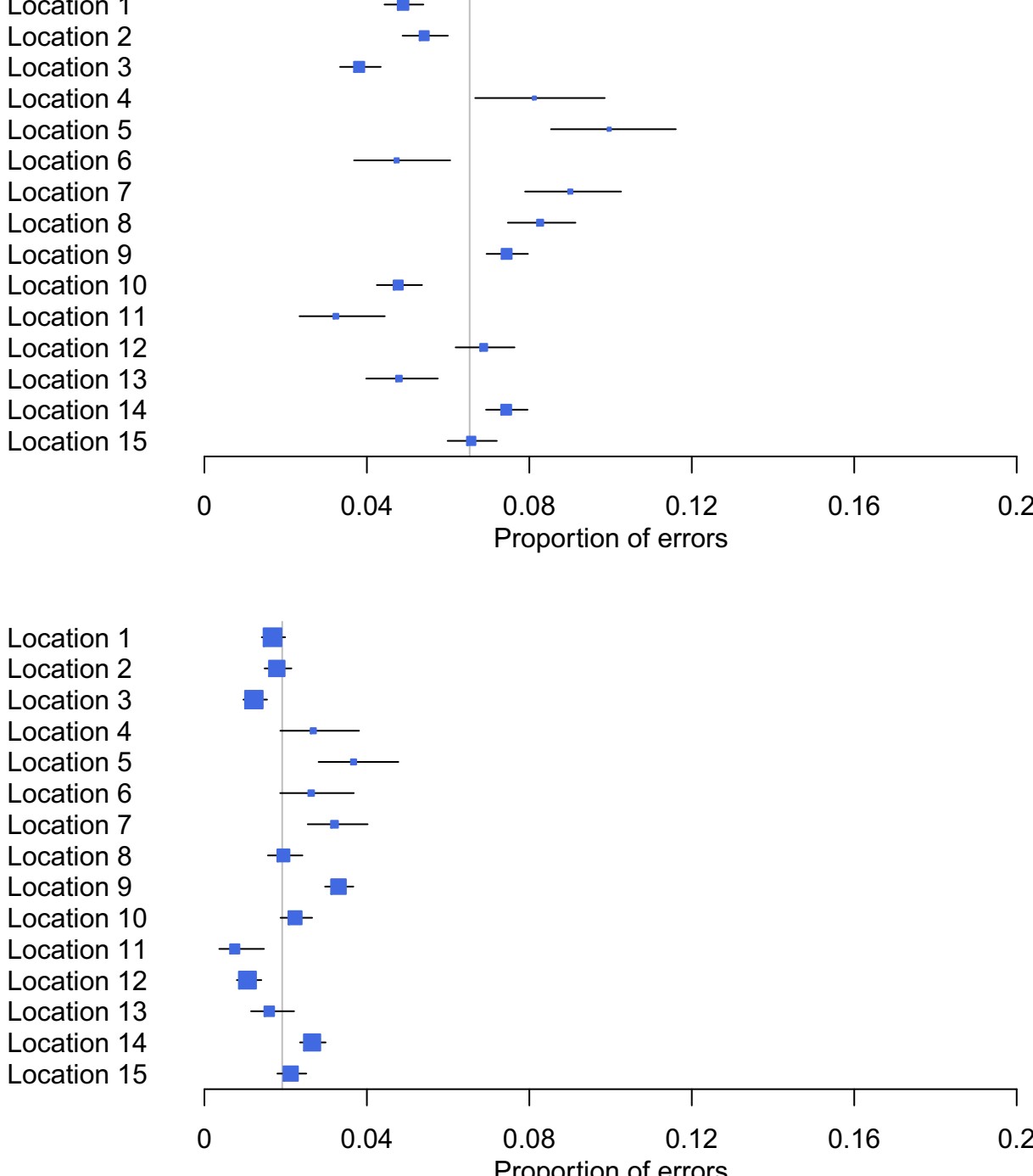

**Fig 2. Estimated rate of all i-STAT errors and user errors (as a proportion of total workload) by location.** Each line shows the estimate of the proportion in a particular location (blue square) and 95% confidence interval (horizontal bars). The grey vertical line shows the estimated proportion across all locations.

Staff experienced problems with both underfilling and overfilling the POCT cartridges. They have found it difficult to deliver the correct amount and are aware that this can lead to error, meaning they are unable to get a result:

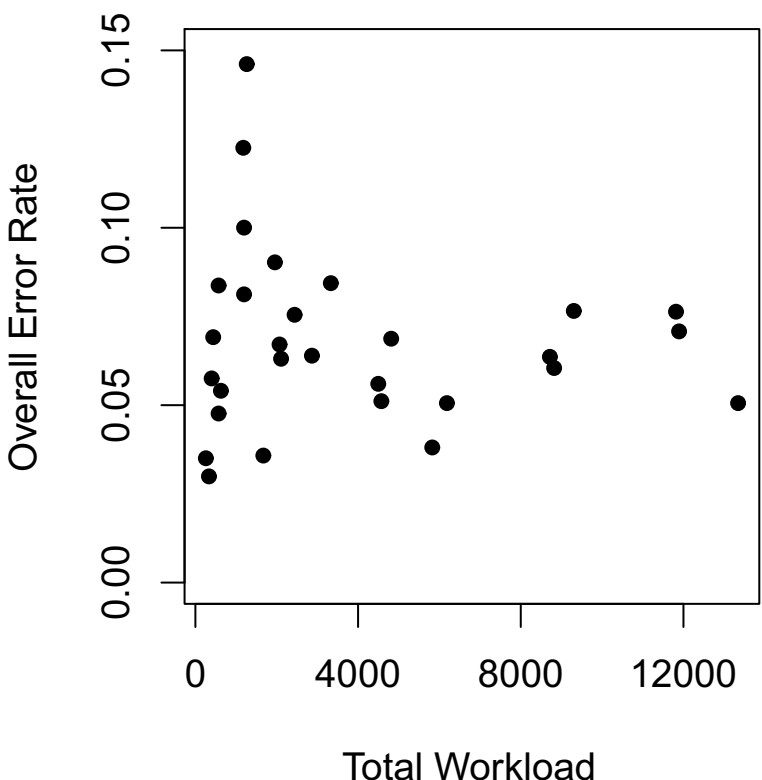

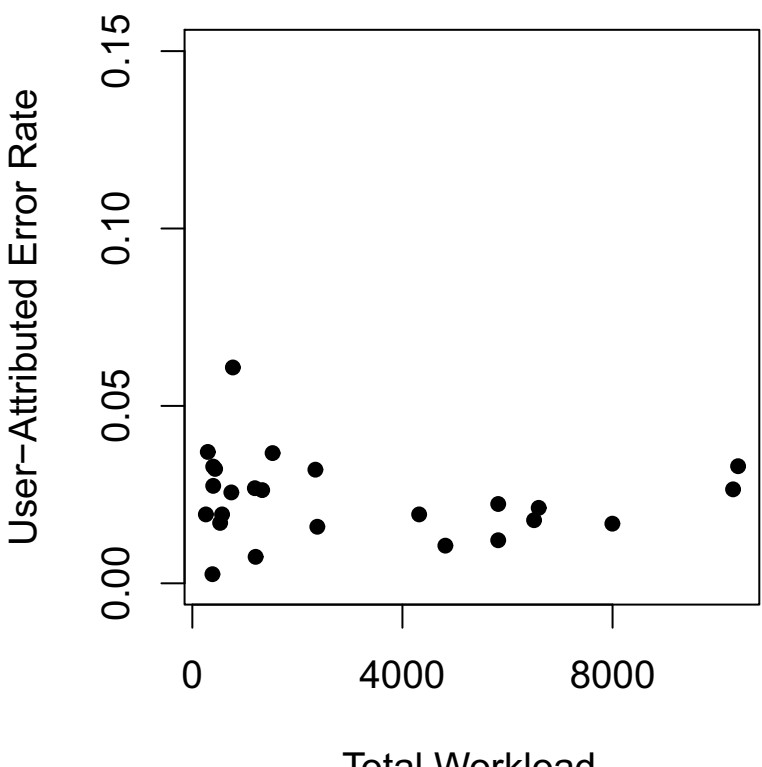

**Fig 3. Relationship between i-STAT error rate and total workload.** All errors (upper panel) and user-attributed errors post-February 2017 (lower panel). Each point represents a different location. Locations with fewer than 200 total uses are excluded to reduce the effect of small sample variability.

"A drop at a time and it's such a tiny, tiny, it's the size of a pipette, it's literally one or two drops, you can't overfill it so you've also got to allow to pull the cap over, use the press stud sort of thing to close it because then it will shoot up, if you put too much in you won't get a result, if you don't put enough in. . ." [Interview 1, HCA, secondary analysis]

Patients attending these ambulatory care settings were often older people living with frailty, and staff reported that this could make the blood sampling process difficult:

"Most of our patients are really difficult to bleed anyway because they're peripherally shut down or, you know, they've got really sort of scarred veins and so you're kind of going in tiny little veins that easily blow and things and so that's just the type of patients we tend to get unfortunately." [Interview 3, Nurse, secondary analysis]

For this reason, some blood samples obtained were very small, if blood could be obtained at all. Using small samples for POCT was seen as problematic by staff as they perceived that this produced error and affected accuracy:

"Or if it's hard to bleed, if you don't get a good flow then the results are not going to be accurate and I think that was the potassium levels, I think that can be a big one, they can be really high which is a concern and sometimes the doctors have said can you get me another specimen because I'm not convinced that that is accurate so we've had to do it again and sure enough, the potassium levels have been lower and so it's obviously the way the blood was taken or the length of time before it got in to the machine so you've got to understand the importance of doing it properly. . ." [Interview 1, HCA, secondary analysis]

Similarly, they perceived that taking a long time to draw the blood could produce inaccurate results:

"I've learnt that if it is totally abnormal and I've had a difficult time getting a blood, or the tourniquet's been on too long or something like that, you know, then it's not going to be accurate." [Interview 1, HCA, secondary analysis]

**Table 4. Total frequencies of error code categories, i-STAT Alinity.**

| Error category | Frequency | Likely user error | Total frequency (%) |
|---|---|---|---|
| Cause 3 codes (Cartridge rejected) | 40 | ✓ | 95 (5.2%) |
| Cause 6 codes (Excess blood) | 18 | ✓ | |
| Cause 7 codes (Cartridge rejected) | 14 | ✓ | |
| Cause 8 codes (Insufficient blood) | 16 | ✓ | |
| Cause 9 codes (Sample rejected) | 7 | ✓ | |
| Cause 15 codes (Cartridge rejected) | 4 | | 15 (0.8%) |
| Codes of unspecified cause | 11 | | |
| Process error codes | 58 | | 58 (3.1%) |
| TOTAL | 168 | | 168 (9.2%) |

Refer to S1 Appendix for error category descriptors.

**Table 5. Total frequencies of error code categories, Afinion.**

| Error code | Frequency | Likely user error | TOTAL |
|---|---|---|---|
| 201 (Insufficient volume) | 35 | ✓ | 50 |
| 203 (Wrong sample) | 6 | ✓ | |
| 208 (Test cartridge previously used) | 9 | ✓ | |
| 101, 102 (Hematocrit error) | 7 | | 44 |
| 213, 214 (Cartridge or analyzer failure) | 6 | | |
| 302 (Analyzer failure) | 31 | | |
| 404 (Operator ID error) | 82 | | 85 |
| MT_16002 (Barcode error) | 3 | | |
| TOTAL | 179 | | 179 |

Refer to S1 Appendix for error category descriptors.

Staff stressed the importance of being able to process blood taken for POCT as quickly as possible, to avoid the blood sample clotting. In the community-based ambulatory care setting, the POCT devices were located close to where blood sampling took place but at busy times there could delays in using them:

> "But I've had it where I've gone in to do my blood and then someone has come in with their blood and they're like Oh I didn't realise, and I'm like I did say I'm doing it, you know, and kind of it's about communicating, so if it's busy it's difficult because you kind of, you might have three patients maybe you're trying to do bloods and stuff on." [Interview 3, Nurse, secondary analysis]

**Staff actions to minimise pre-analytical error.** Staff in both ambulatory care settings regarded the filling of the i-STAT cartridges as the trickiest part of the POCT process. To fill the cartridge, drops of blood are released from the POCT syringe into it. Staff in the community ambulatory care setting explained that when POCT was first introduced a number of cartridges had had to be discarded or were failures because they were incorrectly filled:

> ". . .we wasted an awful lot of cartridges, we were overfilling them, underfilling them, making a real mess, you know, and not getting results, it would just say insufficient blood or whatever" [Interview 1, HCA, secondary analysis]

They have dealt with this difficulty mainly through practice and learning by doing:

> "I've seen people overfill cartridges before and you know they're not going to work but then people learn from that then sort of say "Oh it's overfilled" and then you don't fill it that much again and you kind of have to learn yourself a little bit I think, I don't know." [Interview 3, Nurse, secondary analysis]

> "I think it's just getting used to how much blood to put on and if you do slightly too much, it's a bit kind of accurate how much to put on, but it was fine." [Interview 6, Nurse, secondary analysis]

Staff have also developed their own ways of reducing the messiness of the process:

"The nurse took a piece of kitchen roll out of the dispenser on the wall, put it down on the work surface, put the cartridge on it, and put the blood in the cartridge. . ." [Field notes 8, secondary analysis]

and ensuring that the cartridges are correctly filled:

". . .after the nurse had put some drops of blood on the cartridge, she dabbed the excess off with the corner of the kitchen roll (I've not seen anyone dab the excess off before). . .About blotting the excess blood–she said this was not mentioned in training. If too much blood is sucked up it doesn't work. . .After having some tests not work because too much blood was sucked up, she's started blotting the excess blood. No one told her to do this. She thinks she's seen a couple of other nurses do it; she thinks she saw another nurse do it first." [Field notes 8, secondary analysis]

Staff explained that filling cartridges is an aspect of POCT in which new staff members are given training:

"I asked how she learned to use POCT–she said when she worked here full time previously, she had training in it when she first started. You have to have training, then someone watches you do it for the first few times to make sure you're doing it properly." [Field notes 8, secondary analysis]

"Mainly the procedure, mainly the getting accurate blood results, accurate blood samples and filling the cartridges, that, that was; because even now if we get new nurses, bank nurses or something like that and we show them how to use it, you've got to be really careful how to put those drops of blood in . . ." [Interview 1, HCA, secondary analysis]

During informal training in the hospital-based ambulatory care setting, staff also needed to practise to achieve the necessary control:

"The training to use POC testing is now done by staff in the setting. They get new staff to practise using water in the syringe. They then try to release one drop onto tissue, two drops and so on. They think that control gets better with practice and that practice is important." [Field notes 2, follow-up ethnography]

In the community-based setting staff explained that expired cartridges were used for training purposes:

"There are a few cartridges that have, say, expired that they can practise putting the blood onto, but they won't, you can't actually put those in the machine and get the results 'cos they've got the expired date and it won't let it carry on, but at least you can practise putting the little drops of blood on, yeah." [Interview 6, Nurse, secondary analysis]

Where staff had experienced difficulty in obtaining a sample, they reported that this introduced the possibility of getting air bubbles in the syringe which could negatively affect the POCT process. To overcome this, before filling the cartridge they discarded the first few drops of blood, where the sample was large enough to be able to do this:

"[The HCA] said you're meant to put the first few drops in the waste bucket, before putting drops in the cartridge (I've seen a nurse do this before), but there wasn't enough blood to be

able to do that, so the HCA put it straight into the cartridge. The reason for putting the first drops in the bucket is to stop any air bubbles getting into the cartridge which would stop it reading." [Field notes 6, secondary analysis]

In the community-based setting, the POCT devices were in demand and staff members worked closely together, often in pairs, with one member of staff taking the blood sample and the other staff member taking it immediately to the POCT device, to ensure that the sample was tested as quickly as possible.

POCT, however, is less in demand in the hospital-based ambulatory care setting as clinicians have access to laboratory results within about an hour. At particularly busy times staff can use alternative POCT devices located elsewhere in the hospital but still close by.

**Staff perceptions of error rates.** Staff in both settings were aware of the potential for error in the process and tried to actively reduce the possibility of these occurring. In both settings, blood sampling and POCT was carried out mainly by health care assistants (HCA) but also by nurses. All staff who were involved were familiar with the processes involved and were conducting them multiple times during their working day.

During the field work, there was only one instance observed of an error recorded by a POCT device:

"One of the nurses called me to the clinical room as (s)he had taken some blood from a patient and was about to use the POCT device. (S)he was preparing to do a CHEM8. After discarding the first few drops of blood into the empty cartridge packet (s)he filled a CHEM8 cartridge. (S)he said she was getting rid of air as this might prevent the test working. About a minute or so after (s)he had put the cartridge in, the POCT device gave the error message 'Unable to position sample'. (S)he immediately repeated the process with another cartridge using the same blood sample, which worked second time around. (S)he said she did not know what might have gone wrong with the test and did not say anything about what (s)he thought the error code might mean. (S)he said that the cartridge seldom needed to be done again." [Field notes 3, follow-up ethnography]

Among other staff, there was also a perception that the amount of error occurring was low:

"While observing a POC test being processed in the clinical room, a senior member of staff began to talk about the wastage of cartridges. (S)he estimated that only about one in 50 cartridges was wasted and that this was sometimes due to the wrong amount of blood being inserted in the cartridge. (S)he also mentioned that cartridges were sometimes wasted if the POCT device was not functioning properly." [Field notes 2, follow-up ethnography]

## Discussion

The rate and nature of pre-analytical error when performing POC tests in acute clinical settings has not previously been widely reported. In the period for which the most reliable retrospective data were available, our study estimates the user-attributable error rate using the i-STAT device in a large acute hospital trust to be 2.3%, roughly stable over time, and with overfilling or underfilling the cartridge being the most commonly identified reasons for device-reported error messages. Our conclusions relating to the i-STAT Alinity (estimated user error rate 5.2%) and the Afinion (9.0%) are less certain because of the considerably lower usage of these devices.

These results are consistent with those from the qualitative component of our study, which found that hospital staff saw filling the cartridge as the most difficult aspect of POCT, but that

this rarely resulted in machine error and errors could often be resolved simply by repeating the test with a new cartridge. In our prospective ethnographic field work, we observed only one instance of POC device-recorded error, and this was of a non-serious nature and was promptly resolved. The ethnographic project did however highlight the importance of training, both formal and informal, in ensuring that cartridges were filled correctly to minimise the possibility of pre-analytical errors occurring.

As the focus of our work was on the assessment of pre-analytical error, it is important to realise that these errors do not necessarily directly impact on test results, treatment decisions and patient outcomes, and nor does our estimated error rate cover all types of error that may occur in the 'Total Testing Process' [27]. In most cases, we would expect the types of error our study assesses to be resolved by repeating the test, although there may be impact on resource use through the wastage of a cartridge or the time required to obtain a result. For any particular patient, these factors are unlikely to be substantial as the rate of pre-analytical errors is low, but they may accumulate in busy clinical environments, where hundreds of POC tests are performed daily.

This study has some limitations. Although the quantitative analysis used a complete sample of data that had been recorded automatically by POCT devices at the time of operation, some data had to be excluded because of concerns over inaccurate recording during some periods. Our classification of error codes directly uses the descriptions given in the manufacturers' manuals, but some codes allow for multiple causes and could not definitively be assigned as user errors. This was particularly relevant to the first thirteen months of the study period, when i-STAT error data were recorded as broad categories rather than as specific numerical codes. Our results only relate to the POCT devices already used in the clinical environment considered in our study, and cannot be extrapolated to other analyzers.

More generally, our estimate of error rates only considers test attempts that provoked an error message from the POCT device. It was not possible to assess whether an incorrectly filled cartridge or poor quality sample might have given an inaccurate and therefore clinically misleading test result if the device did not initiate an error message, and this may be affected by the site of sampling or the timing when the test was performed [28]. Additionally, failed attempts in which the operator discarded a cartridge without inserting it into the machine would not be recorded, and this may have introduced some inaccuracy into our estimate of error rates. We had intended to investigate this by cross-classifying ordered stock, test results, and recorded errors, but decided this would be unreliable as departments might vary in the amount of stock retained in reserve at any one time. Additionally, there may be situations in which a cartridge might be discarded without an error having occurred, for example if it were used for practice or training.

The qualitative work was constrained by the time required to observe a sufficient number of test attempts to draw meaningful conclusions. It was not possible to observe all test cartridges and devices, although for most of these the steps required to take blood and fill the cartridge would be similar as i-STAT and Alinity cartridges are identical in design. As the overall error rate was low, we are reliant on staff interviews for information about errors with potentially more damaging consequences. These suggested that these errors would be uncommon, even in care settings such as those used in this study, where difficult or complex cases are frequently encountered. Interviews did however suggest that some difficulties may arise in taking blood from certain patient groups and that filling cartridges was not always easy and required practice. The dependability of these interviews is supported by the fact that one participant estimated the rate of cartridge wastage to be one in fifty, very close to our quantitative estimate. This highlights the importance of appropriate training before undertaking to use blood POCT devices, and this may also include ongoing peer support and additional practice in transferring

blood to the cartridge. Error rates in different locations did not appear to be strongly associated with total test use. This was unexpected but could be explained by location specific factors that we were unable to measure, such as the experience of individual operators. An interesting extra research question would be to investigate whether new operators have higher error rates or improve with practice.

The lack of a consistently-applied taxonomy for describing errors in POCT is a potential barrier to work in this area. We chose to follow that used by Meier & Jones [19] to categorise errors falling into the pre-analytical phase of blood sample collection and suggest that future published reports use a similar classification to allow direct comparisons of error rates in different settings to be made.

In summary, future evaluations of the effect of introducing POCT devices into clinical practice should recognise the need to assess pre-analytical error. Although in this study we conclude that the effect of pre-analytical error is unlikely to be large, this may depend on the device, the setting and the level of expertise of the operators, and so should not be automatically assumed in general. In particular, future work would benefit from an assessment of the effectiveness and impact of training.

## Supporting information

**S1 Appendix. Tables of i-STAT, i-STAT Alinity and Afinion error codes and categories.**
(DOCX)

**S1 Graph. i-STAT total monthly workload and error count.** Monthly workload (solid line) and error count (dotted line) over time (upper panel); enlarged view of error count over time (middle panel); error rate (error count divided by workload) over time (lower panel).
(EPS)

**S1 Data. Data used in quantitative analysis.**
(XLSX)

## Acknowledgments

We gratefully acknowledge the assistance of Tim James for help with service evaluation registration and Alex Novak for facilitating access and providing introduction to the emergency settings where observation took place.

## Author Contributions

**Conceptualization:** Thomas R. Fanshawe, Margaret Glogowska, Philip J. Turner, Gail N. Hayward.

**Formal analysis:** Thomas R. Fanshawe, Margaret Glogowska, George Edwards.

**Funding acquisition:** Philip J. Turner, Gail N. Hayward.

**Investigation:** Thomas R. Fanshawe, Margaret Glogowska, Philip J. Turner, Ian Smith, Rosie Steele, Caroline Croxson.

**Resources:** Ian Smith, Rosie Steele, Jordan S. T. Bowen.

**Software:** Thomas R. Fanshawe, George Edwards.

**Supervision:** Thomas R. Fanshawe, Philip J. Turner, Gail N. Hayward.

**Visualization:** Thomas R. Fanshawe, George Edwards.

**Writing – original draft:** Thomas R. Fanshawe, Margaret Glogowska.

**Writing – review & editing:** Thomas R. Fanshawe, Margaret Glogowska, George Edwards, Philip J. Turner, Ian Smith, Rosie Steele, Caroline Croxson, Jordan S. T. Bowen, Gail N. Hayward.

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
