## [Decision Letter · Decision Letter 0]

24 Dec 2019

PONE-D-19-29304

Pre-analytical error for point of care venous blood testing in acute ambulatory settings : a mixed methods service evaluation

PLOS ONE

Dear Dr Fanshawe,

Thank you for submitting your manuscript to PLOS ONE. After careful consideration, we feel that it has merit but does not fully meet PLOS ONE’s publication criteria as it currently stands. Therefore, we invite you to submit a revised version of the manuscript that addresses the points raised during the review process.

In addition to the few issues raised by the reviewers, please reconsider revising the title to clearly state that the paper is on the effect of pre-analytic handling  on three POCT instrument errors. One of the most common pro-analytic errors insufficient patient ID.

We would appreciate receiving your revised manuscript by Feb 07 2020 11:59PM. To enhance the reproducibility of your results, we recommend that if applicable you deposit your laboratory protocols in protocols.io, where a protocol can be assigned its own identifier (DOI) such that it can be cited independently in the future. For instructions see: http://journals.plos.org/plosone/s/submission-guidelines#loc-laboratory-protocols

We look forward to receiving your revised manuscript.

Kind regards,

Pal Bela Szecsi, M.D. D.M.Sci.

Academic Editor

PLOS ONE

Journal Requirements:

This research was funded by the National Institute for Health Research (NIHR) Community Healthcare MedTech and In Vitro Diagnostics Co-operative at Oxford Health NHS Foundation Trust. The views expressed are those of the authors and not necessarily those of the NHS, the NIHR or the Department of Health and Social Care. The project was also supported by a grant from Becton, Dickinson and Company. The funders had no role in study design, data collection and analysis, decision to publish, or preparation of the manuscript.

We note that you received funding from a commercial source: Becton, Dickinson and Company.

Reviewers' comments:

Reviewer's Responses to Questions

**Comments to the Author**

1. Is the manuscript technically sound, and do the data support the conclusions?

Reviewer #1: Yes

Reviewer #2: Yes

2. Has the statistical analysis been performed appropriately and rigorously? 

Reviewer #1: N/A

Reviewer #2: N/A

3. Have the authors made all data underlying the findings in their manuscript fully available?

Reviewer #1: Yes

Reviewer #2: Yes

4. Is the manuscript presented in an intelligible fashion and written in standard English?

Reviewer #1: Yes

Reviewer #2: Yes

5. Review Comments to the Author

Reviewer #1: The study consider the pre—analytical errors recorded with the use of three specific devices, both in an hospital setting and community in ambulatory care. The authors consider the errors registered by the instruments and in particular the errors attributable to the operators.

The errors in the pre-analytical phase include many aspects, in addition to those that can be registered by the analytical system: the choice of the right timing with respect to the clinical conditions (for example for blood gas, you could quote Auvet doi: 10.1186/s13613-016-0152-6 ), the site of the sampling, the instruments used for venipuncture, the any dilution or contamination of the blood drawn, the sample hemolysis. Despite this, the study provides new data on the safety of POCT systems. This limitation must be discussed.

Another limitation of the study consists in the typology of errors that are strictly related to the specific type of instrument considered. These results cannot be reported to other types of analyzers.

In the qualitative study the secondary analysis is based on limited number of cases.

The "Staff awareness of factors", “staff actions” and “staff perception” chapters (lines 329 - 512) present individual anecdotal cases that could be placed in a section of additional materials rather than in the text of the paper. Alternatively, the proposed reports could be summarized.

In the conclusions a positive judgment is expressed regarding the frequency of errors attributable to the operators. But it is not known what the impact of these errors on care was, as the authors themselves claim, therefore the statement appears arbitrary and must be more prudent. In absolute terms the frequency is actually high, for example if we estimate it with the sigma metric.

Figure 3 shows little relevant data that can be briefly presented in the text.

Reviewer #2: This is a well written manuscript with focus on a topic for which there are very few published studies.

I only have one comment. The authors found that there was no difference in error rates between locations performing high and low volume POCT. I would have expected the error rate to be lower in areas performing high volume testing. Could the authors speculate or comment on why they saw no difference?

6. PLOS authors have the option to publish the peer review history of their article (what does this mean?). If published, this will include your full peer review and any attached files.

Reviewer #1: No

Reviewer #2: No

---

## [Author Response · Author response to Decision Letter 0]

9 Jan 2020

Response to reviewers

*Editor:

*In addition to the few issues raised by the reviewers, please reconsider revising the title to clearly state that the paper is on the effect of pre-analytic handling on three POCT instrument errors.

We agree and have modified the title to reflect this:

Pre-analytical error for three point of care venous blood testing platforms in acute ambulatory settings : a mixed methods service evaluation

*One of the most common pro-analytic errors insufficient patient ID.

This was true of the Afinion but this was likely to be only a technical result in accessing the device (quickly resolved by the operator) and so not likely to be an operator error. As such, although we thought it appropriate to report this in results tables we do not feel that it warrants more prominence than currently.

*Reviewer #1: The study consider the pre—analytical errors recorded with the use of three specific devices, both in an hospital setting and community in ambulatory care. The authors consider the errors registered by the instruments and in particular the errors attributable to the operators.

The errors in the pre-analytical phase include many aspects, in addition to those that can be registered by the analytical system: the choice of the right timing with respect to the clinical conditions (for example for blood gas, you could quote Auvet doi: 10.1186/s13613-016-0152-6 ), the site of the sampling, the instruments used for venipuncture, the any dilution or contamination of the blood drawn, the sample hemolysis. Despite this, the study provides new data on the safety of POCT systems. This limitation must be discussed.

Thank you – we agree that this should be mentioned and we have added to the relevant part of the Discussion: “It was not possible to assess whether an incorrectly filled cartridge or poor quality sample might have given an inaccurate and therefore clinically misleading test result if the device did not initiate an error message, and this may be affected by the site of sampling or the timing when the test was performed [28]” (citing the paper suggested).

*Another limitation of the study consists in the typology of errors that are strictly related to the specific type of instrument considered. These results cannot be reported to other types of analyzers.

We have added a sentence to the limitations in the Discussion: “Our results only relate to the POCT devices already used in the clinical environment considered in our study, and cannot be extrapolated to other analyzers.”

*In the qualitative study the secondary analysis is based on limited number of cases. The "Staff awareness of factors", “staff actions” and “staff perception” chapters (lines 329 - 512) present individual anecdotal cases that could be placed in a section of additional materials rather than in the text of the paper. Alternatively, the proposed reports could be summarized.

In qualitative research, emphasis is placed on gaining rich, deep information from a relatively small number of cases, where each has substantial experience of, and involvement in, the phenomenon under investigation (in this case, POC testing). The secondary analysis strand is based on data collected in a qualitative case study of one ambulatory care unit published elsewhere (Jones C, Glogowska M, Locock L, Lasserson D. Embedding new technologies in practice – a normalization process theory study of point of care testing. BMC Health Surv Res. 2016;16:591) which encompassed 14 episodes of ethnographic observation (approximately 60 hours) and 7 semi-structured interviews with staff (6 of which were audio-recorded and transcribed). In a similar way, the prospective ethnographic strand (which took place in another ambulatory care unit), represented 15 hours of observation. 

For a qualitative study, this represents a considerable dataset, which was able to provide both a description of the context in which the POC testing had been implemented and a detailed interpretation of how usual practice around POC testing took place.

As regards the “Staff awareness of factors”, “staff actions” and “staff perception” sections, the material here are rigorously collected and robustly analysed data from the secondary analysis and the prospective ethnographic strands of the qualitative component. The qualitative component was part of the original mixed methods design which aimed at producing a more comprehensive assessment of preanalytical error in POC testing in this setting. Where quotations are included, it is because they are reflective of the views of more than just the one participant to whom they are attributed and are not simply ‘anecdotal cases’. Thus, we regard the qualitative material as playing an important role in aiding our understanding, and providing some explanation of, the quantitative findings and do not think it should be moved into additional materials.

*In the conclusions a positive judgment is expressed regarding the frequency of errors attributable to the operators. But it is not known what the impact of these errors on care was, as the authors themselves claim, therefore the statement appears arbitrary and must be more prudent. In absolute terms the frequency is actually high, for example if we estimate it with the sigma metric.

We agree that our study was unable to assess the impact on patient care (this was not the objective). We have amended the Conclusions of the Abstract as a result: “In the observed settings, the rate of errors attributable to operators of the primary point of care device was less than 1 in 40. In some cases, errors may lead to a small increase in resource use or time required so adequate staff training is necessary to prevent adverse impact on patient care.”

*Figure 3 shows little relevant data that can be briefly presented in the text.

We have moved this figure to a Supporting Information file (S2 Graph) and shortened the text.

*Reviewer #2: This is a well written manuscript with focus on a topic for which there are very few published studies.

I only have one comment. The authors found that there was no difference in error rates between locations performing high and low volume POCT. I would have expected the error rate to be lower in areas performing high volume testing. Could the authors speculate or comment on why they saw no difference?

Thank you – we had also anticipated that the error rate might be lower in areas performing high volume testing but were limited in the amount of location-specific information that could be collected to determine factors that might influence error rates at different locations. We have amended the relevant part of the Discussion to now read: “Error rates in different locations did not appear to be strongly associated with total test use. This was unexpected but could be explained by location specific factors that we were unable to measure, such as the experience of individual operators. An interesting extra research question would be to investigate whether new operators have higher error rates or improve with practice.”

---

## [Decision Letter · Decision Letter 1]

22 Jan 2020

Pre-analytical error for three point of care venous blood testing platforms in acute ambulatory settings : a mixed methods service evaluation

PONE-D-19-29304R1

Dear Dr. Fanshawe,

We are pleased to inform you that your manuscript has been judged scientifically suitable for publication and will be formally accepted for publication once it complies with all outstanding technical requirements.

With kind regards,

Pal Bela Szecsi, M.D. D.M.Sci.

Academic Editor

PLOS ONE

Additional Editor Comments (optional):

Reviewers' comments:

Reviewer's Responses to Questions

**Comments to the Author**

1. If the authors have adequately addressed your comments raised in a previous round of review and you feel that this manuscript is now acceptable for publication, you may indicate that here to bypass the “Comments to the Author” section, enter your conflict of interest statement in the “Confidential to Editor” section, and submit your "Accept" recommendation.

Reviewer #1: All comments have been addressed

Reviewer #2: All comments have been addressed

2. Is the manuscript technically sound, and do the data support the conclusions?

Reviewer #1: Yes

Reviewer #2: Yes

3. Has the statistical analysis been performed appropriately and rigorously? 

Reviewer #1: N/A

Reviewer #2: N/A

4. Have the authors made all data underlying the findings in their manuscript fully available?

Reviewer #1: Yes

Reviewer #2: Yes

5. Is the manuscript presented in an intelligible fashion and written in standard English?

Reviewer #1: (No Response)

Reviewer #2: Yes

6. Review Comments to the Author

Reviewer #1: The paper was corrected as requested. However, I believe that research can be deepened with a new study with respect to clinical risk (for example through interviews, questionnaires, event analysis) and with respect to patient safety. Also the process of patient identification sometimes is a critical issue. But this is only a my suggestion.

Reviewer #2: The authors have sufficiently addressed the concerns I raised. This manuscript is well written and addresses an important topic.

7. PLOS authors have the option to publish the peer review history of their article (what does this mean?). If published, this will include your full peer review and any attached files.

Reviewer #1: No

Reviewer #2: No

---

## [Editor Report · Acceptance letter]

27 Jan 2020

PONE-D-19-29304R1 

Pre-analytical error for three point of care venous blood testing platforms in acute ambulatory settings : a mixed methods service evaluation 

Dear Dr. Fanshawe:

I am pleased to inform you that your manuscript has been deemed suitable for publication in PLOS ONE. Congratulations! Your manuscript is now with our production department. 

With kind regards,

on behalf of

Dr. Pal Bela Szecsi 

Academic Editor

PLOS ONE